# Reproductive Ability Disparity in the Pacific Whiteleg Shrimp (*Penaeus vannamei*): Insights from Ovarian Cellular and Molecular Levels

**DOI:** 10.3390/biology13040218

**Published:** 2024-03-27

**Authors:** Jianchun Zhang, Jie Kong, Jiawang Cao, Ping Dai, Baolong Chen, Jian Tan, Xianhong Meng, Kun Luo, Qiang Fu, Peiming Wei, Sheng Luan, Juan Sui

**Affiliations:** 1National Key Laboratory of Mariculture Biobreeding and Sustainable Goods, Yellow Sea Fisheries Research Institute, Chinese Academy of Fishery Sciences, Qingdao 266071, China; zhangjianchun227@163.com (J.Z.); kongjie@ysfri.ac.cn (J.K.); caojw@ysfri.ac.cn (J.C.); daiping54@163.com (P.D.); 1320040091@163.com (B.C.); tannjian@163.com (J.T.); mengxianhong@ysfri.ac.cn (X.M.); luokun@ysfri.ac.cn (K.L.); oucfuq@163.com (Q.F.); 2Laboratory for Marine Fisheries Science and Food Production Processes, Laoshan Laboratory, Qingdao 266237, China; 3BLUP Aquabreed Co., Ltd., Weifang 261312, China; jakmail@163.com

**Keywords:** *Penaeus vannamei*, fecundity, mitochondria, EHF, PRDM9

## Abstract

**Simple Summary:**

In order to investigate the molecular mechanisms responsible for the variations in reproductive capacity among *Penaeus vannamei* shrimp females, we analyzed individuals exhibiting significantly different spawning abilities. Subcellular variances in ovarian composition were observed in these distinct cohorts of females across different maturity stages. Additionally, we conducted a transcriptional-level screening of candidate genes and associated pathways implicated in ovarian maturation. Furthermore, we assessed the expression profiles of the divergent candidate genes EHF and PRDM9 at different stages of ovarian development, along with their cellular localization within the ovaries. These findings are anticipated to provide valuable insights into the molecular mechanisms underlying the differences in reproductive capacity among shrimp females.

**Abstract:**

The Pacific whiteleg shrimp (*Penaeus vannamei*) is a highly significant species in shrimp aquaculture. In the production of shrimp larvae, noticeable variations in the reproductive capacity among female individuals have been observed. Some females experience slow gonadal development, resulting in the inability to spawn, while others undergo multiple maturations and contribute to the majority of larval supply. Despite numerous studies that have been conducted on the regulatory mechanisms of ovarian development in shrimp, the factors contributing to the differences in reproductive capacity among females remain unclear. To elucidate the underlying mechanisms, this study examined the differences in the ovarian characteristics between high and low reproductive bulks at different maturity stages, focusing on the cellular and molecular levels. Transmission electron microscopy analysis revealed that the abundance of the endoplasmic reticulum, ribosomes, mitochondria, and mitochondrial cristae in oocytes of high reproductive bulk was significantly higher than that of the low reproductive bulk in the early stages of ovarian maturation (stages I and II). As the ovaries progressed to late-stage maturation (stages III and IV), differences in the internal structures of oocytes between females with different reproductive capacities gradually diminished. Transcriptome analysis identified differentially expressed genes (DEGs) related to the mitochondria between two groups, suggesting that energy production processes might play a crucial role in the observed variations in ovary development. The expression levels of the ETS homology factor (EHF) and PRDI-BF1 and RIZ homology domain containing 9 (PRDM9), which were significantly different between the two groups, were compared using qRT-PCR in individuals at different stages of ovarian maturation. The results showed a significantly higher expression of the EHF gene in the ovaries of high reproductive bulk at the II and IV maturity stages compared to the low reproductive bulk, while almost no expression was detected in the eyestalk tissue of the high reproductive bulk. The PRDM9 gene was exclusively expressed in ovarian tissue, with significantly higher expression in the ovaries of the high reproductive bulk at the four maturity stages compared to the low reproductive bulk. Fluorescence in situ hybridization further compared the expression patterns of EHF and PRDM9 in the ovaries of individuals with different fertility levels, with both genes showing stronger positive signals in the high reproductive bulk at the four ovarian stages. These findings not only contribute to our understanding of the regulatory mechanisms involved in shrimp ovarian development, but also provide valuable insights for the cultivation of new varieties aimed at improving shrimp fecundity.

## 1. Introduction

The Pacific whiteleg shrimp (*Penaeus vannamei*) is the highest-yielding shrimp species in the world due to its rapid growth, wide salt tolerance, strong disease resistance, and high protein content. In 2021, the global aquaculture production of shrimp reached 6.3 million tons [1]. The production of high-quality seedlings is crucial for the sustainable development of the shrimp aquaculture industry. High-quality shrimp broodstock is a prerequisite for the sustainable production of seedlings. Over the past few decades, the technique of eyestalk ablation (ESA) has been extensively employed in the artificial breeding of decapod crustaceans to stimulate ovarian maturation and reduce the spawning interval [2,3]. By eliminating the X-organ sinus gland complex (XO–SG) situated in the eyestalk ganglion, this technique effectively eliminated the source of hormones that hinder ovary development [3,4,5]. However, significant differences in the spawning frequency of female shrimp have been observed in the actual production process. In previous studies, the percentage of female shrimp that did not reach maturity has ranged from 14% to 51% [6,7]. This situation results in significant resource wastage and has a severe impact on the economic benefits of shrimp farming operations.

The maturation of crustacean ovaries represents a highly energy-intensive process. Research indicates that this complex process is regulated by stimulatory factors originating from the X-organ sinus gland complex (XO–SG) [8,9,10], a vital neuroendocrine organ located within the eyestalk of crustaceans, playing a pivotal role in ovary development and maturation. Key hormones including crustacean hyperglycemic hormone (CHH), molt-inhibiting hormone (MIH), vitellogenesis/gonad-inhibiting hormone (VIH/GIH), and mandibular organ-inhibiting hormone (MOIH) are synthesized and released by the XO–SG to collectively regulate physiological and biochemical processes such as growth, reproduction, and ecdysone secretion [11,12,13,14]. Additionally, genes such as *vasa* and the cathepsin family also play significant roles in ovarian development [15,16,17]. Moreover, various cellular components within the oocyte, including the perioval space inclusions, endoplasmic reticulum, and mitochondria, may be involved in yolk granule formation [18]. Despite numerous studies on the mechanisms of ovarian maturation in crustaceans, the intrinsic factors responsible for variations in shrimp spawning frequency remain unclear.

In previous studies, we selected females exhibiting marked variations in spawning frequency as our research subjects. Utilizing selective sweep analysis, we pinpointed 145 genomic regions associated with ovarian development. Subsequently, we identified 121 candidate genes within these regions and delineated the expression patterns of certain genes across various tissues and stages of ovarian maturation in female shrimp with different spawning frequencies [19,20]. Moreover, we performed histological examinations on the ovaries of shrimp females with different spawning frequencies, uncovering the crucial processes such as yolk granule formation and accumulation, as well as cortical rod formation, which occurred at a faster pace in high fertility female shrimp compared to the low fertility shrimp [20]. Nevertheless, the molecular mechanisms underlying the variations in reproductive capacity among shrimp female individuals remain unclear. This study concentrated on females with notable differences in spawning frequency as the research subjects. We analyzed ultrastructural differences between high and low spawning females at the same mature stage. Furthermore, we analyzed the transcriptome of mature ovaries to identify potential key genes associated with maturation rate. Additionally, we examined the gene expression patterns of two candidate genes, EHF and PRDM9, across different ovarian maturity stages and tissues, and conducted cellular localization studies on ovaries. These findings not only enhance our understanding of the regulatory mechanism governing ovarian development in shrimp, but also provide valuable insights for the cultivation of new varieties aimed at enhancing shrimp fecundity.

## 2. Materials and Methods

### 2.1. Animals

The experimental shrimp were sourced from BLUP Aquabreed Co., Ltd. (Weifang, Shandong Province, China). From August to September 2021, 632 11-month-old females and 600 males from 79 families underwent large-scale maturation induction to produce future generations. Females were raised in six cement ponds (16 m^2^) at an average density of 6–7 individuals/m^2^. Males from the same family were raised in cement ponds (3 m^2^) at a density of 2–3 individuals/m^2^. The water was changed once a day, replacing 80% of the water volume each time. The temperature of the female shrimp pond was 28 °C, and the temperature of the male shrimp pond was 27 °C. Both ponds had a salinity of 31, pH 8.0~8.2, ammonia nitrogen ≤ 0.5 mg/L, and nitrite ≤ 0.05 mg/L.

### 2.2. Induction of Ovary Development and Spawning Frequency Statistics

The shrimp candidate parents were provided with a feeding schedule, consisting of squid at 7 am and 5 pm, along with commercial parent shrimp fortified feed at 10:30 am, 2 pm, and 7 pm daily. To synchronize the ovarian development, female shrimp underwent unilateral eyestalk ablation after one month of maturation promotion. Simultaneously, a plastic ring with a four-digit code was placed on another eyestalk as a unique identifier for each female. After a 10-day recovery period post-eyestalk ablation, family construction was initiated, which lasted for a total of 45 days. Each day, shrimp females at stage IV of ovarian development, indicative of imminent spawning, were individually selected. After recording the ring code number, the mature female was individually introduced into specified male shrimp ponds for natural mating, according to a pre-designed mating plan. Following the culmination of family construction, the spawning frequency of 632 shrimp females was recorded. Females with no signs of ovary maturation were categorized as exhibiting low reproductive bulk, while those with four or more ovary maturation events were classified as possessing high reproductive bulk. Each group comprised 40 females and was placed in 3 m^2^ cement ponds for sustained nutritional fortification. Each pool contained 10–15 females, and their ovarian development was monitored every day. Female ovarian development can be divided into four stages, based on ovarian volume, color, size, shape, and structure [21]. This study used the same ovarian staging method, which involved observing the ovaries with a flashlight.

### 2.3. Sample Collection

Electron microscope sample: Shrimp females at different ovary maturation stages were selected from both high and low reproductive bulks. Subsequently, they were placed on ice for anesthesia. The intact fresh ovarian tissues were promptly dissected into 1–2 mm^3^ pieces. These tissue samples were then immersed in electron microscope fixation solution (Serxicebio, G1102, Wuhan, China) and fixed at room temperature (24–25 °C) for 2 h in a light-free environment. The active ingredient of the fixative is 2.5% glutaraldehyde, 0.1 M phosphate buffer as solvent, pH 7.0–7.5 at 25 °C. Then, the fixed ovarian tissues were stored at 4 °C. The fixative to tissue ratio was maintained at not less than 10:1.

Transcriptome sequencing sample: Twelve high reproductive bulk and twelve low reproductive bulk were divided into three parallel groups, each containing four individuals. The groups were labeled as KGT1, KGT2, KGT3 for the high reproductive bulk, and KDT1, KDT2, and KDT3 for the low reproductive bulk. The shrimp samples were swiftly placed on ice and anesthetized. Ovarian tissues were excised and stored in liquid nitrogen.

Fluorescence in situ hybridization and qRT-PCR samples: Both the high and low reproductive bulks were put on ice and anesthetized. Fresh and intact ovarian tissues at stages I–IV were swiftly immersed in a fixative solution (Serxicebio, G1113, Wuhan, China) at 4 °C for subsequent fluorescence in situ hybridization experiments. The fixative to tissue ratio was maintained at no less than 10:1. The active ingredient of the fixative is 4% paraformaldehyde, with 0.1 M phosphate buffer as the solvent, pH 7.0–7.5 at 25 °C, treated with DEPC to effectively inactivate RNase in the solution. Meanwhile, three shrimp females were randomly selected from each stage of both the high and low reproductive bulks. Ovary, eyestalk, and hepatopancreas tissues were collected, stored in liquid nitrogen, and subsequently transferred to a −80 °C refrigerator for future RNA extraction and qRT-PCR validation.

### 2.4. Transmission Electron Microscopy Sample Preparation

The fixed tissue for electron microscopy was subjected to multiple washes with a 0.1 M phosphate buffer solution (pH 7.4). It was then treated with 1% osmium acid for 2 h at room temperature in the dark, followed by an additional wash with 0.1 M phosphate buffer solution (pH 7.4). Subsequently, the tissue underwent dehydration using a gradient of alcohol (30%, 50%, 70%, 80%, 90%, 95%, and 100%) for 20 min at each concentration. It was then transferred twice to anhydrous acetone, with each transfer lasting 15 min. Following this, the tissue was embedded in an embedding agent and left overnight in an oven set at 37 °C, followed by polymerization for 48 h in an oven at 60 °C. Once the temperature naturally returned to room temperature (25 ± 0.5 °C), the embedding plate was removed from the oven. It was then sectioned using an ultrathin microtome and subsequently stained. The resulting sections were observed and photographed under a transmission electron microscope.

### 2.5. RNA-Seq and Bioinformatic Analysis

#### 2.5.1. RNA-Seq of Ovarian Samples

The ovarian samples were subjected to total RNA extraction using a Trizol kit (Invitrogen, Carlsbad, CA, USA), following the manufacturer’s protocol. Subsequently, the extracted RNA underwent quality assessment using an Agilent 2100 bioanalyzer (Agilent Technologies, Palo Alto, CA, USA) and agarose gel electrophoresis. mRNA containing a polyA tail was enriched using Oligo (dT) magnetic beads and then fragmented via ultrasonication. Reverse transcription was employed to synthesize the first cDNA strand, followed by the synthesis of the second cDNA strand. The purified double-stranded cDNA underwent terminal repair, A tail addition, and ligation to a sequencing adapter. AMPure XP beads were used to select cDNA fragments of approximately 200 bp, which were then subjected to PCR amplification. The PCR products were purified again using AMPure XP beads to yield the final library, which was sequenced on an Illumina HiSeq 2500 platform (Gene Denovo Biotechnology Co., Guangzhou, China).

#### 2.5.2. Data Quality Control and Sequence Alignment

After preprocessing the raw data, high-quality clean reads were obtained by filtering out low-quality data using fastp [22]. Ribosome reads were removed from the analysis using Bowtie2 [23], and the remaining data were aligned to the reference genome of the Pacific whiteleg shrimp using HISAT2 2.1.0.

#### 2.5.3. Expression Level Statistics and Inter-Group Analysis

Following the alignment with HISAT2, transcripts were reconstructed using Stringtie [24], and the expression levels of all the genes in each sample were determined using RSEM [25]. The count data obtained from gene expression analysis were further analyzed using DESeq2 [26]. Differentially expressed genes (DEGs) were identified based on the results of the differential analysis, with a screening threshold of *p* < 0.05 and |log_2_FC| ≥ 1. Gene Ontology (GO) function and KEGG pathway enrichment analyses were performed on the DEGs to identify significantly enriched GO terms and pathways.

### 2.6. qRT-PCR and Statistical Analysis

Total RNA was extracted using an RNA kit (Vazyme, Nanjing, China), and cDNA was synthesized using HiScript III R T SuperMix for qPCR (+gDNA wiper) (Vazyme, Nanjing, China). Subsequently, qRT-PCR was performed using SYBR Green Real-time PCR Master Mix (TOYOBO, Shanghai, China). Primers for qRT-PCR targeting the mRNA sequences of the ETS homologous factor (EHF) (XM_027371458.1), and PRDI-BF1 and RIZ homology domain containing 9 (PRDM9) (XM_027354640.1), were designed using Primer3.0 online, with 18S serving as an internal reference (Table 1). The expression levels of the EHF and PRDM9 genes in different tissues and ovarian stages were analyzed using the 2^−ΔΔCt^ calculation method. Statistical significance was evaluated using the IBM SPSS Statistics 26 software, with a statistical significance level of *p* < 0.05 indicating significant differences, and *p* < 0.01 indicating extremely significant differences.

### 2.7. Fluorescence In Situ Hybridization (FISH)

The fixed ovarian tissue was processed into paraffin sections, followed by proteinase K digestion (20 μg/mL). Prehybridization was carried out by adding a prehybridization solution, which was then removed before adding a hybridization solution containing 500 nM of the probes EHF and PRDM9. Incubation was performed overnight at 40 °C in a thermostat. The sections were washed with 2 × SSC for 10 min at 37 °C, followed by 1 × SSC for 2 × 5 min at 37 °C, and 0.5 × SSC for 20 min at room temperature. After gently drying, 60 μL of preheated probe mixture was added, and the sections were placed horizontally in a wet box at 40 °C for 45 min. During this process, 50 mL of 2 × SSC was added to the bottom of the wet box. Subsequently, the hybridization solution was removed, and the sections were washed with preheated 2 × SSC, 1 × SSC, 0.5 × SSC, and 0.1 × SSC for 5 min at 40 °C. The sections were gently dried and incubated with a preheated signal probe hybridization solution (60 μL) in a wet box at 40 °C for 45 min. Again, 50 mL of 2 × SSC was added to the bottom of the wet box. After pouring out the hybridization solution, the sections were washed with preheated 2 × SSC, 1 × SSC, 0.5 × SSC, and 0.1 × SSC for 5 min at 40 °C. DAPI staining solution was added to the sections, incubated for 8 min in the dark, and washed before sealing with an anti-fluorescence quenching sealing agent. The sections were observed, and images were collected under a Nikon upright fluorescence microscope. The SweAMI probe for fluorescence in situ hybridization was synthesized by Wuhan Jinkairui Biological Co., Ltd., Wuhan, China, and the probe sequence is shown in Table 2.

## 3. Results

### 3.1. Maturation Frequency of 632 Female Shrimp in a Breeding Cycle

The maturation frequency of 632 shrimp females during a 45-day production cycle of Pacific whiteleg shrimp is illustrated in Figure 1. The maturation frequency is characterized by the following: 44.62% of females did not mature, 43.83% of females matured one to three times, and only 14% of females matured four times or more.

### 3.2. Comparison of Ovarian Ultrastructure between High and Low Reproductive Female Shrimp

In stage I of ovarian development, there were significant differences in the rough endoplasmic reticulum (RER) between high and low reproductive females. The ovaries of high reproductive females showed abundant long tubular RER structures in the cytoplasm, with a large number of attached ribosome particles (Figure 2a), whereas the ovaries of low reproductive females displayed RER structures with sparse ribosome attachment (Figure 2b). As we progressed to stage II of ovarian development, an increase in organelles within the cytoplasm was evident (Figure 2c,d). Notably, the RER structures in the ovaries of low reproductive females continued to exhibit limited ribosome attachment (Figure 2e). A comparison of the mitochondria between the ovaries of high and low reproductive females revealed numerous elongated cristae in high reproductive females’ ovary mitochondria, whereas the cristae in low reproductive females’ ovary mitochondria were fewer and shorter (Figure 2f). Subsequent to advancement to stage III, both the quantity and length of cristae in the mitochondria of high reproductive females’ ovaries began to increase (Figure 2g). Additionally, the number of RER and attached ribosomes in the ovaries of low reproductive females increased (Figure 2h). Stage IV revealed a substantial presence of spherical dense yolk granules in the ovaries of both high and low reproductive females (Figure 2i,j).

### 3.3. Transcriptome Data

#### 3.3.1. Overview of RNA Sequencing Data

The results of individual transcriptome data filtering are presented in Table 3. A total of 230,932,272 raw reads were obtained from the samples. After a rigorous data filtering process, 271,580,334 clean reads were retained by removing low-quality and short sequences. The average proportion of clean to raw reads was 98.77%. The Q20 quality score exceeded 97.28% and the Q30 quality score surpassed 92.49%. Furthermore, the average GC content was determined to be 49.11%.

The comparative analysis of the reads and reference genomes is presented in Table 4. The examination of the data showed a high alignment rate, surpassing 93%, with an average alignment rate of 93.52%. The alignment rates for multiple comparisons ranged from 27.46% to 29.90%, averaging 27.71%. In contrast, the alignment rates for unique comparisons ranged from 65.47% to 66.12%, with an average rate of 65.81%.

#### 3.3.2. Differential Gene Expression Analysis

The comparative analysis between high and low reproductive bulks revealed a total of 132 differentially expressed genes (DEGs). Among these, 47 genes were upregulated, and 85 genes were downregulated, as depicted in Figure 3 and Appendix A.

#### 3.3.3. GO Enrichment Analysis of DEGs

Based on the GO enrichment results (Figure 4), the high vs. low reproductive bulk exhibited enrichment of 402 DEGs in biological processes, 102 DEGs in molecular functions, and 244 DEGs in cellular components. In the Top 20 bubble of differentially expressed genes’ GO enrichment (Figure 5), the top five significantly enriched categories were as follows: Cellular component assembly involved in morphogenesis (GO: 0010927), Contractile fiber part (GO: 0044449), Actomyosin structure organization (GO: 0031032), Anatomical structure formation involved in morphogenesis (GO: 0048646), and Contractile fiber (GO: 0043292).

#### 3.3.4. KEGG Annotation of DEGs

In the comparison of ovarian samples between the high and low reproductive bulks, a total of 40 DEGs were annotated into 36 signaling pathways, as illustrated in Figure 6. The KEGG enrichment analysis revealed the top five enriched KEGG pathways to be the Adipocytokine signaling pathway (ko04920), Influenza A (ko05164), Pathogenic *Escherichia coli* infection (ko05130), Valine, leucine, and isoleucine biosynthesis (ko00290) and Thyroid hormone signaling pathway (ko04919), as depicted in Figure 7.

### 3.4. Analysis of EHF and PRDM9 Genes in DEGs

#### 3.4.1. Expression Pattern of EHF Gene and Localization of Ovarian Cells at Different Mature Stages

In ovarian tissue, the expression pattern of the EHF gene demonstrated a sequence of decline, followed by an increase, and then another decline in accordance with ovarian development (Figure 8). The low reproductive bulk exhibited notably low expression levels of EHF genes in stages II, III, and IV. Specifically, the expression level of EHF genes in stage I was approximately 59 times higher than that of stage II, and 22 times higher than that of stage III. The expression of EHF genes also exhibited a distinct pattern when comparing the high and low reproductive bulks. While no significant distinction between the two groups was noted in stage I, the high reproductive bulk displayed significantly higher EHF gene expression in stages II, III, and IV compared to the low reproductive bulk (*p* < 0.05). However, in the eyestalk tissue, a contrasting scenario emerged, where the expression level of the EHF gene in the low reproductive bulk exceeded that of the high reproductive bulk, with the EHF gene essentially absent in the latter. In the hepatopancreas tissue across stages I–IV, the high reproductive bulk exhibited a higher expression level of the EHF gene compared to the low reproductive bulk, with this discrepancy being particularly pronounced in stages I, II, and III (*p* < 0.01).

To depict the ovarian localization and expression of EHF across the different ovary developmental stages, fluorescence in situ hybridization was employed. As illustrated in Figure 9, the positive signal (red) mainly appeared in the cytoplasm of oocytes. In the ovaries of both the high and low reproductive bulks, a significant expression signal was observed in stage I, while the signal weakened in stages II–III and intensified in stage IV. Additionally, the positive signal’s intensity was notably stronger in the oocytes of the high reproductive bulk compared to the low reproductive bulk.

#### 3.4.2. Expression Pattern Analysis of PRDM9 Gene and Localization of Ovarian Cells at Different Mature Stages

The qRT-PCR results showed that the expression of the PRDM9 gene was restricted to ovarian tissues, with no expression in the eyestalk and hepatopancreas. Within the ovarian tissues of the high reproductive bulk (Figure 10), the expression of the PRDM9 gene displayed a distinct pattern characterized by an initial decline, followed by an increase, and eventually another decline. The lowest expression level was observed in stage II, while the highest expression level was recorded in stage III.

In the low reproductive bulk, the PRDM9 gene’s expression demonstrated an ascending trend, followed by a subsequent decline. The peak expression of the PRDM9 gene occurred during stage II, subsequently decreasing during stage III, where it represented only 1/24th of the level observed in stage II. The expression of PRDM9 during stage IV remained exceedingly low, almost negligible. In the comparative analysis between the high and low reproductive bulks, the high reproductive bulk significantly surpassed the low reproductive bulk in both stage I and stage III (*p* < 0.01). Furthermore, during stage IV, the high reproductive bulk exhibited significantly higher expression compared to the low reproductive bulk (*p* < 0.05).

To assess the cellular localization and expression of PRDM9 in the ovaries across various developmental stages, fluorescence in situ hybridization was employed (Figure 11). The positive signal (green) predominantly appeared in the cytoplasm of oocytes. Except for the low reproductive bulk during stage I, affirmative signals were evident in both high and low reproductive bulks during other stages. Particularly noteworthy, the positive signal exhibited increased intensity during stage IV compared to other stages. In stages I, III, and IV, the positive signal’s strength in the high reproductive bulk significantly exceeded that of the low reproductive bulk.

## 4. Discussion

Although both the high and low reproductive bulks reached maturity in the ovaries during the four phases, and electron microscopy results showed that both groups had accumulated a large amount of yolk granules, it can be observed from practical production experience that the ovarian maturation cycle of the low reproductive bulk is longer than that of the high reproductive bulk. Histological observations of ovarian tissues from the high and low reproductive bulks at different stages showed apparent differences between their organelles such as mitochondria, rough endoplasmic reticulum, and ribosomes in the ovary.

The mitochondria in the early stage of ovarian development in the low reproductive bulk showed sparse and short cristae. Meanwhile, transcriptome analysis revealed that some mitochondrial-related genes, including the ATP synthase lipid-binding protein gene, the mitochondrial fission 1 protein-like gene, the iron sulfur cluster assembly 1 homolog, and mitochondrial-like gene, were differentially expressed between the high and the low reproductive bulks. Mitochondria exist in the cytoplasm of eukaryotic cells and serve as energy factories, generating ATP to maintain cellular metabolic activity [28]. For the shrimp, the energy-intensive process from ovarian development to maturation, as well as from oocyte development to mature oocytes, requires a large amount of energy. Mitochondria are the main sites of energy production [29], providing the energy required for oocyte maturation, thereby promoting oocyte maturation. Abnormal mitochondrial function in oocytes can cause abnormal meiosis and embryonic development defects in mice [30,31]. Ca^2+^ plays a signaling role as a second messenger in various cellular life activities, and an abnormal mitochondrial calcium concentration in oocytes can hinder the progression of meiosis [32]. In addition, studies have shown that mitochondria are involved in the formation of yolk granules within oocytes. Although Beams et al. [33] believed that there was no relationship between the mitochondria and the formation of yolk granules in the red claw shrimp, Jiang et al. [34] observed that mitochondria were involved in the formation of yolk granules in the oocytes of *Exopalaemon modestus*. In mammals, the endoplasmic reticulum can synthesize functional proteins through messenger RNA translation and can store and release free calcium ions in the cytoplasm. The homeostasis of the endoplasmic reticulum is crucial for the normal development and maturation of oocytes, as well as follicular development and oocyte maturation [35]. Studies in mice have reported interactions between the endoplasmic reticulum and mitochondria in oocytes [36].Therefore, the difference in mitochondria is probably a main reason for the different ovarian maturation cycles of the two bulks. Protein synthesis and degradation of ingested substances are energy consumption processes, indicating that the differences in the rough endoplasmic reticulum and ribosome may be a consequence of different morphological structures of mitochondria. Our recent study found that 145 selective sweep regions were identified in the genome between both high and low reproductive bulks, with some key genes enriched in the serotonin (5-HT) pathway [19]. Our previous verification showed that the expression of the *PLCβ4* gene in the 5-HT pathway was higher in high reproductive bulk than in low reproductive bulk at different stages of ovarian development [20]. Studies show that there is an intricate relationship between the 5-HT and mitochondria, and that specific 5-HT signaling regulates mitochondrial biogenesis. In cultured mouse cortical neurons, signaling of the 5-HT2A receptor caused peroxisome proliferation, activating the receptor γ cofactor 1-α, promoting ATP production and increasing mitochondrial DNA quality and antioxidant capacity [37]. In studies of rodent kidney injury, spinal cord injury, and Parkinson’s disease, 5-HT1F receptor agonists can induce mitochondrial biogenesis [37,38]. Stimulation of the 5-HT7 receptors induced ATP production in the impaired rat model of Rett syndrome in the brain, revealing the potential role of 5-HT in the regulation of mitochondrial bioenergy [39]. In human breast cancer, 5-HT signaling-induced mitochondrial biosynthesis, as well as subsequent changes in cellular metabolism, contributed to an increase in oxidative phosphorylation [40]. In many crustaceans, 5-HT has been reported to regulate oocyte growth and ovarian development [41,42,43,44]. Previous studies have shown that 5-HT regulates ovarian development by affecting the synthesis and release of endocrine factors, such as VIH, MIH, VSH and RPCH, in the eyestalk and nervous system of crustaceans [45,46,47]. Therefore, we speculated that the shrimp neuroendocrine, 5-HT pathway, oocyte mitochondrial metabolism, and other processes jointly affect the ovarian development process through complex regulation. Further investigations need to be carried out to clarify the underlying mechanisms.

Transcriptome analysis and further expression detection also identified some genes that might be related to ovary development. ETS transcription factors are known to play a significant role in animal growth, development, cell proliferation, differentiation, apoptosis, and organ formation. In *Xenopus*, *ets-2* has been shown to be crucial for oocyte development and maturation [48]. *Elg*, another ETS family gene, is crucial for egg development in *Drosophila* [49]. During the development of early human embryos and germ cells, the EHF gene is associated with genome-wide DNA hydroxymethylation (5hmC) processes, which is a pivotal process for normal organismal development [50]. Notably, during stage IV of ovarian development in the low reproductive bulk, the EHF gene was almost completely unexpressed. Fluorescence in situ hybridization results further confirmed the cytoplasmic expression of the EHF gene, with the signal intensity of the stage IV ovarian EHF gene in the high reproductive bulk being notably stronger than that of the low reproductive bulk. These data suggest that the EHF gene might play an important role in the ovarian development and oocyte maturation of shrimp.

PRDM9 is 1 of the 16 identified PRDM methyltransferases which play vital roles in cell differentiation, embryonic development, and the onset of specific diseases [51]. PRDM proteins feature an N-terminal PR domain associated with the SET methyltransferase domain, along with multiple zinc finger structures facilitating sequence-specific DNA binding and protein–protein interactions [52]. PRDMs function as direct histone methyltransferases or recruit histone-modifying enzymes to target promoters, influencing developmental signaling pathways and cellular state transitions [53]. In mice, PRDM9 is mainly expressed in the early stages of germ cell meiosis, and abnormal function of this gene can cause infertility in organisms [54]. Knockout of the PRDM9 gene in both female and male mice leads to infertility because the deletion of the PRDM9 gene impairs the repair pathway for double-stranded DNA breaks, subsequently affecting the pairing of homologous chromosomes and causing impaired sex body formation [55]. Low expression of the PRDM9 gene in the testes may be linked to male infertility in cattle [56]. In mice, both female and male mice were rendered infertile by the knockout of PRDM14, another PRDM member specifically expressed during germ cell development [57]. In the present study, PRDM9 was exclusively expressed in the ovaries of the adult shrimp and its expression was nearly absent in the stage IV ovarian tissue of the low reproductive bulk when compared to that of the high reproductive bulk, which was confirmed by the fluorescence in situ hybridization analysis. The data suggest that the PRDM9 gene might play a pivotal role in the ovarian maturation and oocyte development of shrimp.

## 5. Conclusions

In this study, we analyzed the molecular mechanisms of the reproductive differences between female shrimp. According to the results of the transmission electron microscopy, compared with the low- and high-fecundity shrimps, the organelles in the ovary are not well developed, especially the long tubular rough endoplasmic reticulum, mitochondria, and ribosomes. Through transcriptome analysis of the ovaries of high and low reproductive shrimp, a total of 132 differentially expressed genes were screened. Among the differentially expressed genes, EHF and PRDM9 were studied. The qRT-PCR and fluorescence in situ hybridization results revealed the potential significance of EHF and PRDM9 genes in the process of ovarian development and maturation in shrimp.

## Figures and Tables

**Figure 1 biology-13-00218-f001:**
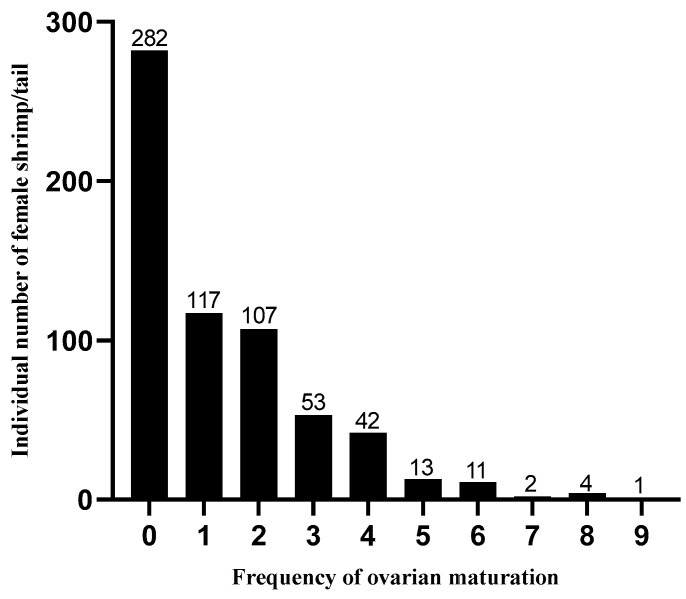
Maturation frequency of 632 shrimp females in a production cycle (45 d) of the *P. vannamei*.

**Figure 2 biology-13-00218-f002:**
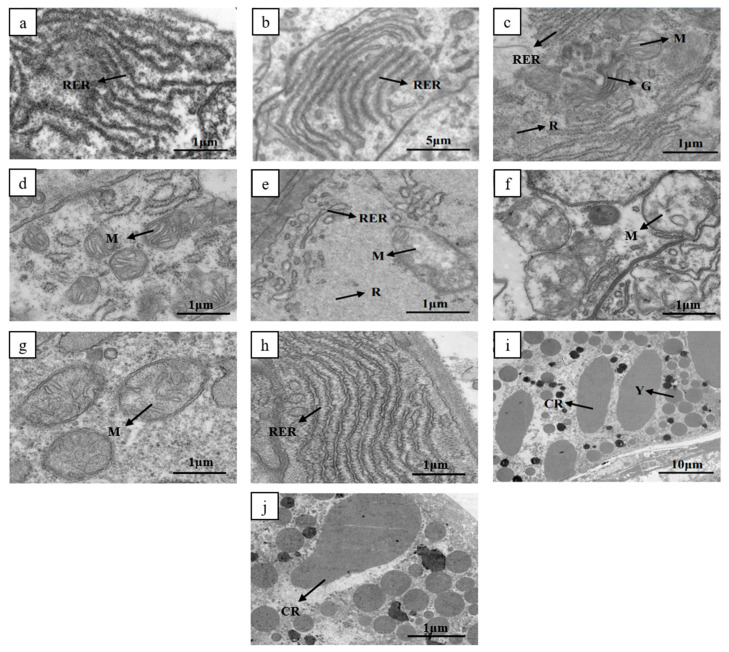
Transmission electron microscopy of ovaries with high and low reproductive bulks in stages I–IV. (**a**) Long tubular rough endoplasmic reticulum in oocytes of high reproductive bulk in stage I; (**b**) long tubular rough endoplasmic reticulum in oocytes of low reproductive bulk in stage I; (**c**) organelles within the cytoplasm of high reproductive bulk in stage II; (**d**) mitochondria of high reproductive bulk in stage II; (**e**) organelles within the cytoplasm of low reproductive bulk in stage II; (**f**) mitochondria of low reproductive bulk in stage II; (**g**) mitochondria of low reproductive bulk in stage III; (**h**) long tubular rough endoplasmic reticulum in oocytes of low reproductive bulk in stage III; (**i**,**j**) intracytoplasmic yolk granules and cortical rods of high and low reproductive bulks in stage IV. RER: Long tubular rough endoplasmic reticulum; M: Mitochondria; R: Ribosome; G: Golgi apparatus; Y: yolk granules; CR: Cortical rod.

**Figure 3 biology-13-00218-f003:**
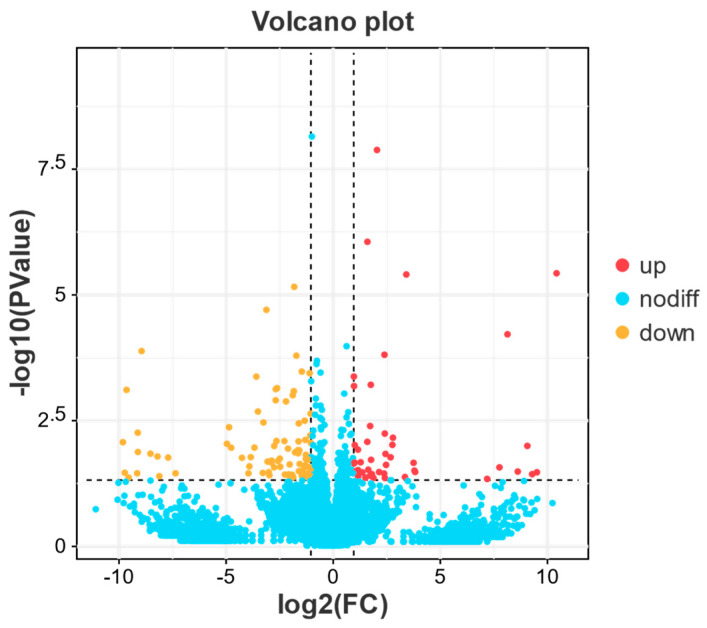
Differential gene volcano map of high vs. low reproductive bulk. The abscissa is the log2 value of the fold difference; the ordinate is the −log10 value of the *p*−value or FDR value; the red scatter points represent upregulated differentially expressed genes; the yellow scatter points represent downregulated differentially expressed genes; the blue scatter points represent genes that do not meet the threshold screening.

**Figure 4 biology-13-00218-f004:**
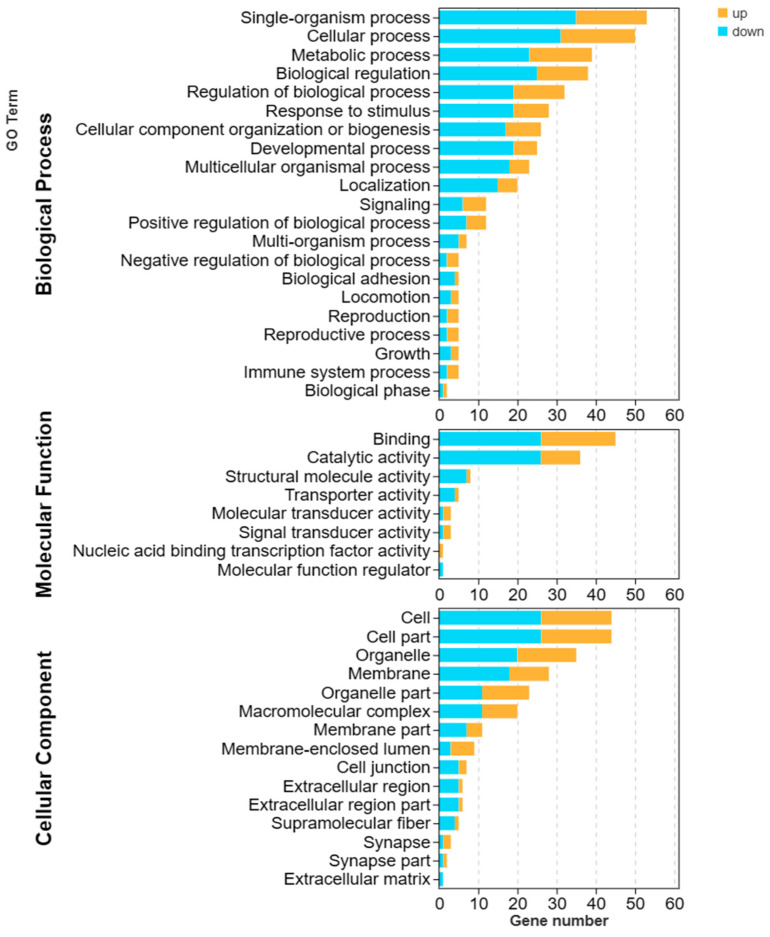
GO annotation classification statistics of high vs. low reproductive bulk. The abscissa is the secondary GO term, and the ordinate is the number of genes in the term. Orange indicates upregulation, and blue indicates downregulation. *p* < 0.05 and |log_2_FC| > 1.

**Figure 5 biology-13-00218-f005:**
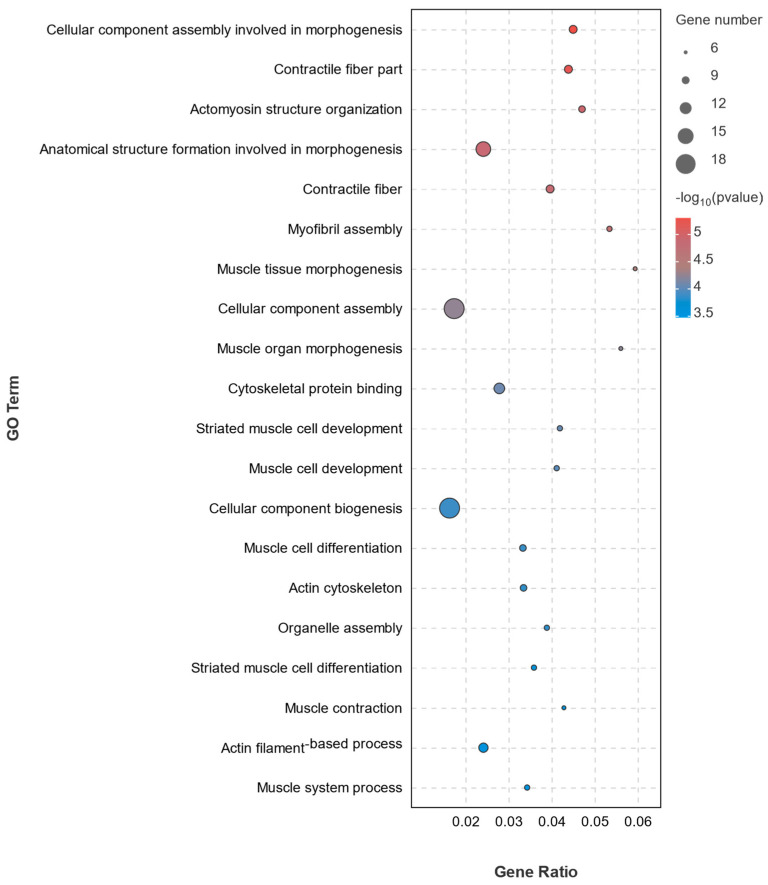
GO enrichment of differentially expressed genes in the Top 20. The abscissa is Gene Ratio, and the ordinate is GO Term. The size of the bubbles indicates the number of differences in enrichment into the GO term. The color of the bubbles indicates significant enrichment in the GO term.

**Figure 6 biology-13-00218-f006:**
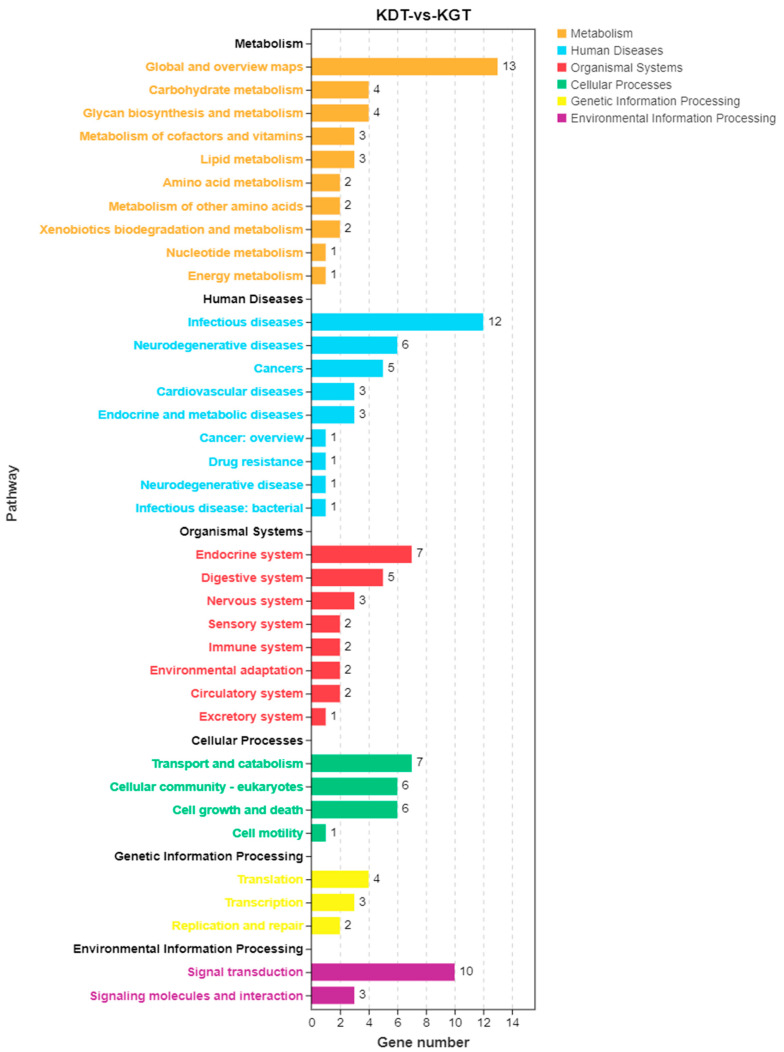
KEGG signaling pathway enrichment analysis of differentially expressed genes in ovarian tissue of high vs. low reproductive bulk. The abscissa is the number of genes, the ordinate is the pathway name, each column represents a pathway, and the column height represents the number of genes contained in the pathway. Different colors represent different first level classifications.

**Figure 7 biology-13-00218-f007:**
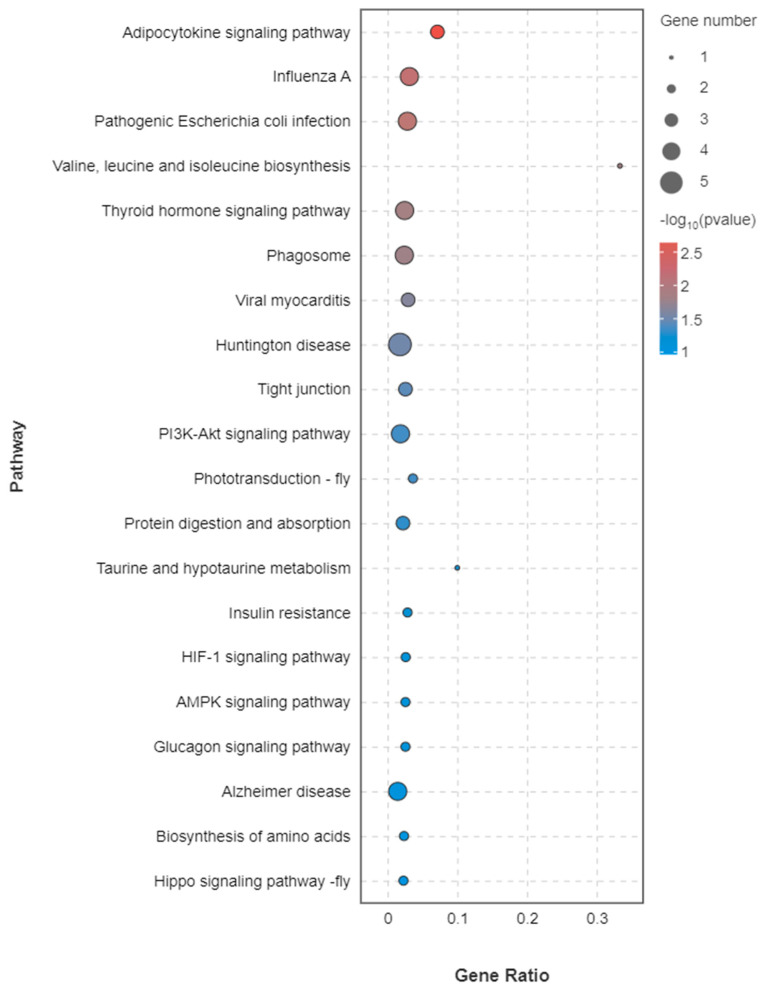
KEGG enrichment of differentially expressed genes in the Top 20. The abscissa is gene ratio, and the ordinate is the pathway name. The size of the bubble indicates the number of differentially expressed genes enriched in the pathway. The color of the bubble indicates the significance of enrichment in the pathway.

**Figure 8 biology-13-00218-f008:**
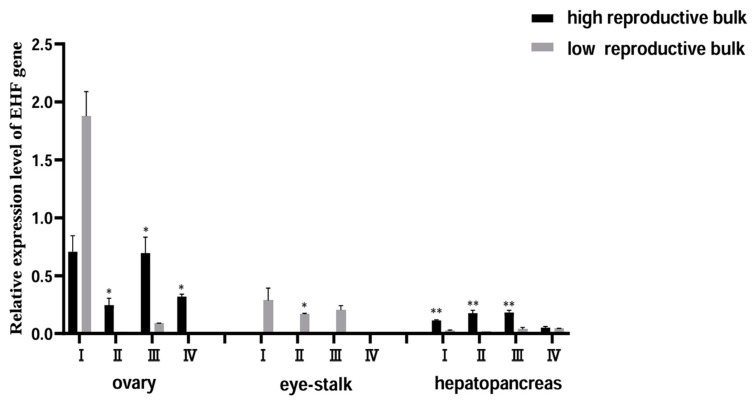
The expression pattern of the EHF gene in different tissues of high and low reproductive bulks at different ovary stages. * represents a significant difference between the high and low reproductive bulks at the same stage (*p* < 0.05), ** represents a highly significant difference between the high and low reproductive bulks at the same stage (*p* < 0.01). I–IV represents different stages of ovarian development.

**Figure 9 biology-13-00218-f009:**
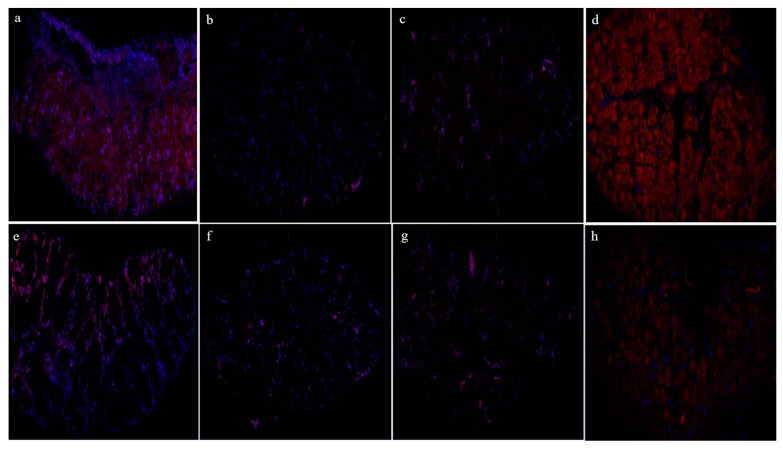
Fluorescence in situ hybridization results of the EHF gene in the I–IV ovaries of the high and low reproductive bulks. (**a**–**d**) Fluorescence in situ hybridization results of I–IV stage ovaries in the high reproductive bulk; (**e**–**h**) Fluorescence in situ hybridization results of I–IV stage ovaries in the low reproductive bulk.

**Figure 10 biology-13-00218-f010:**
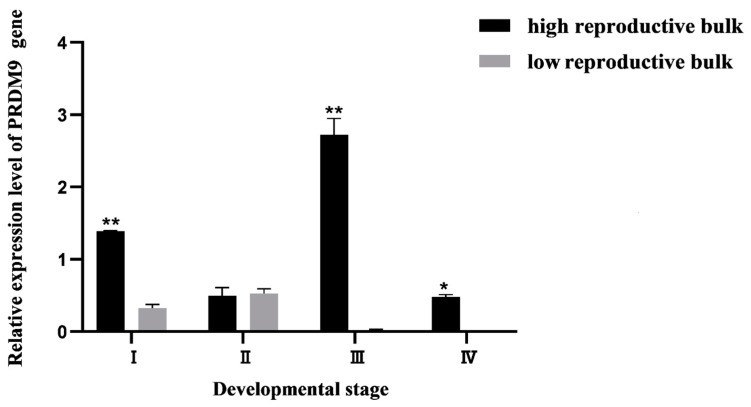
The expression pattern of PRDM9 gene in ovary of high and low reproductive bulks at different ovary stages. * represents significant difference between the high and low reproductive bulks at the same stage (*p* < 0.05), ** represents highly significant difference between the high and low reproductive bulks at the same stage (*p* < 0.01). I–IV represents different stages of ovarian development.

**Figure 11 biology-13-00218-f011:**
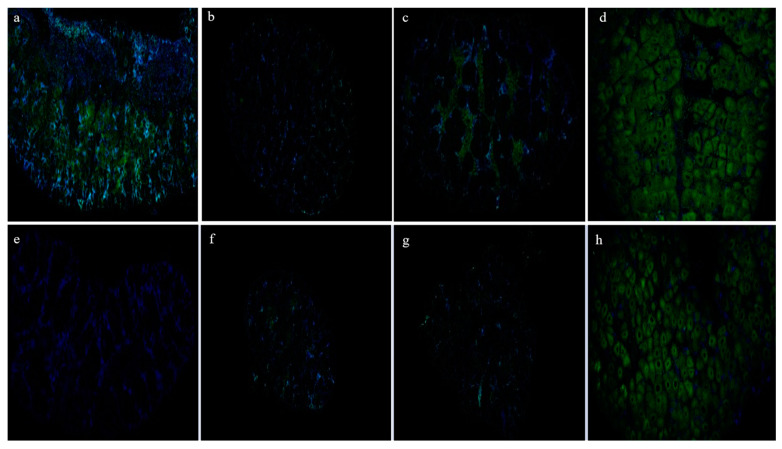
Fluorescence in situ hybridization results of the PRDM9 gene in the I–IV ovaries of the high and low reproductive bulks. (**a**–**d**) Fluorescence in situ hybridization results of I–IV stage ovaries in the high reproductive bulk; (**e**–**h**) Fluorescence in situ hybridization results of I–IV stage ovaries in the low reproductive bulk.

**Table 1 biology-13-00218-t001:** Primer sequences for qRT-PCR.

Primer Name	Sequence (5′-3′)	Fragment Length (bp)	TM/°C	Reference
18S-F	TATACGCTAGTGGAGCTGGAA	147	59	Shi et al. (2016) [27]
18S-R	GGGGAGGTAGTGACGAAAAAT			
EHF-F	CAAGGTCTATGGCGATGGAGAATGG	115	60	The present study
EHF-R	TGGTAGTGATTGGAAGGCGAGGTAG			
PRDM9-F	GCCAGGAGTGAGAACTACGATGATG	142	59	The present study
PRDM9-R	CTCCACTTGACCTGTCGGCTATTG			

**Table 2 biology-13-00218-t002:** Fluorescence in situ hybridization probe sequences.

Gene Name	Probe Sequence
EHF	5′-GATGCCGATTTGTCTTTTTTTGGTC-3′
	5′-ACCATCTGGCAACACTCCTGTCCT-3′
	5′-GAAACCCGAAGAACACTGTAATCCG-3′
PRDM9	5′-GTTGCGTCGCTTCCTCTGTTGTT-3′
	5′-CCGCCTACGCTTCCATTTTTGT-3
	5′-CACTCTGCCTTCCCCTTATGACC-3′

**Table 3 biology-13-00218-t003:** Statistical table for data filtering of reads and reference genome alignment.

Group	Raw Data	Clean Data	AF-Q20 (%)	AF-Q30 (%)	AF-GC
KDT1	46,520,442	46,288,750 (99.50%)	6,715,262,029 (97.44%)	6,399,880,267 (92.87%)	3,395,451,947 (49.27%)
KDT2	41,907,276	41,722,290 (99.56%)	6,040,934,649 (97.28%)	5,742,949,875 (92.49%)	3,063,680,490 (49.34%)
KDT3	44,090,614	43,907,530 (99.58%)	6,372,805,328 (97.53%)	6,077,816,927 (93.02%)	3,238,788,915 (49.57%)
KGT1	47,324,512	47,103,908 (99.53%)	6,831,597,125 (97.54%)	6,521,213,357 (93.11%)	3,482,699,644 (49.73%)
KGT2	47,507,396	47,263,324 (99.49%)	6,862,474,739 (97.58%)	6,550,690,414 (93.15%)	3,453,633,923 (49.11%)
KGT3	45,450,430	45,294,532 (99.66%)	6,585,079,758 (97.59%)	6,288,860,139 (93.20%)	3,334,162,581 (49.41%)

**Table 4 biology-13-00218-t004:** Reference statistics of reads and reference genome alignment.

Group	Total	Unmapped (%)	Unique_ Mapped (%)	Multiple_ Mapped (%)	Total_ Mapped (%)
KDT1	46,076,744	2,956,233 (6.42%)	30,466,203 (66.12%)	12,654,308 (27.46%)	43,120,511 (93.58%)
KDT2	41,505,912	2,804,291 (6.76%)	27,175,879 (65.47%)	11,525,742 (27.77%)	38,701,621 (93.24%)
KDT3	43,747,662	2,772,509 (6.34%)	28,830,719 (65.90%)	12,144,434 (27.76%)	40,975,153 (93.66%)
KGT1	46,969,794	3,032,178 (6.46%)	30,984,340 (65.97%)	12,953,276 (27.58%)	43,937,616 (93.54%)
KGT2	46,992,062	3,140,999 (6.68%)	30,799,565 (65.54%)	13,051,498 (27.77%)	43,851,063 (93.32%)
KGT3	45,165,458	2,808,346 (6.22%)	29,754,848 (65.88%)	12,602,264 (27.90%)	42,357,112 (93.78%)

## Data Availability

Data are contained within the article and Appendix A.

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
