# Peer review of "Reproductive Ability Disparity in the Pacific Whiteleg Shrimp (Penaeus vannamei): Insights from Ovarian Cellular and Molecular Levels"

_biology, 2024, doi:10.3390/biology13040218_

Round 1

Reviewer 1 Report

Comments and Suggestions for Authors

The article investigated reproductive ability Litopenaeus vannamei broadstocks from ovarian cellular and molecular levels. The manuscript is interesting. The cited references are provided adequate background regarding this topic. The goal of study is clear. The results are completed. I have a few minor suggestions that could enhance the quality of the manuscript.

Abstract

Line 37-40: The obtained  results need to be clearly stated in the article abstract.

 Line 41-43: rewrite the conclusion section.

Material and Method:

 Line 208-210. Have you conducted a normality test?

Discussion:

Discussion part needs to be deeply improved. There is some intriguing data presented by the author, but the discussion is superficial.

However, I recommend that the manuscript be ready for publication after some minor revisions..

Reviewer 2 Report

Comments and Suggestions for Authors

This manuscript as a whole poses a very interesting problem. It consists of studying the reproductive ability of female shrimp Litopenaeus vannamei, namely the mechanisms regulating the development of their ovaries, which underlie the formation of differences in the reproductive ability of females. To elucidate the underlying mechanisms, this study compared ovarian differences between high- and low-fertility female shrimp at different stages of maturity at the cellular and molecular levels.

The potential of this work is good, but it seems to me that the language of presentation needs careful correction and rewriting of the manuscript in many parts.

Litopenaeus vannamei is known as the Pacific white shrimp. I recommend including this common name in the text and subsequently avoiding using the Latin name, replacing it with the common name or simply "shrimp", since no other species have been studied.

Everywhere in text "female shrimps" should be changes as "shrimp females" or "L. vannamei females"

Lines 15-17. Please reconsider as "To investigate the molecular mechanisms underlying the differences in reproductive capacity among shrimp Litopenaeus vannamei females, we studied females with significant different spawning abilities".

Lines 78-79. Please reconsider as "In previous studies, we chose females with significant variations in spawning frequency as research subjects.”

Lines 90-93. Please reconsider as "We analyzed ultrastructural differences between high and low spawning females at the same mature stage. Furthermore, we analyzed the transcriptome of mature ovaries to identify potential key genes associated with maturation rate”.

Lines 102 -107. Please reconsider as "Lines From August to September 2021, 632 11-month-old females and 600 males from 79 families underwent large-scale maturation induction to produce future generations. Females were raised in six cement ponds at an average density of 6-7 individuals per m2. Males from the same family were raised in cement ponds (3 m2) at a density of 2-3 individuals/m2”.

Add area of ponds for females

Lines 124-128. Please reconsider as "Each pool contained 10-15 females, and their ovarian development was monitored every day. Female ovarian development can be divided into four stages based on ovarian volume, color, size, shape, and structure [21]. This study used the same ovarian staging method, which involved observing the ovaries with a flashlight”.

Line 136. Add here that “The active ingredient of the fixative is 2.5% glutaraldehyde, 0.1M phosphate buffer as solvent, pH 7.0-7.5 at 25℃”.

Line 147. Add composition of fixative Serxicebio, G1113, Wuhan, China.

Line 163 “ room temperature” Please provide exact number in ℃.

Lines 214-218. Please reconsider as "The fixed ovarian tissue was processed into paraffin sections, followed by proteinase K digestion (20 μg/ml). Prehybridization was carried out by adding a prehybridization solution, which was then removed before adding a hybridization solution containing 500 nM of the probes EHF and PRDM9. Incubation was performed overnight at 40 °C in a thermostat."

Lines 240-241/ Please check here numbers in percentages. 282 (no matured)  < 287 (1, 2 and 3 maturations) but 45%>44%.

Lines 247-248 “In stage I of ovarian development, there were significant differences in the rough endoplasmic reticulum (RER) between high and low reproductive females. The ovaries of high reproductive females…”

In Figure 4 the font on the X axis is too small and cannot be read.

Line 433. Are you sure that is specific feature for L. vannamei?

The discussion should be expanded.

The discussion should consider the mechanisms regulating the development of their ovaries, which underlie the formation of differences in the reproductive ability of females, since this is the main goal of the work. 

The authors found potential significance of EHF and PRDM9 genes in the process of ovarian development and maturation in studied species of decapods. There are very few comparisons with other animal groups. 

Lines 439 and 448. Add common names of animals.

Add more literature. See useful sources:

Vervoort M, Meulemeester D, Behague J, Kerner P. Evolution of Prdm Genes in Animals: Insights from Comparative Genomics. Mol Biol Evol. 2016 Mar;33(3):679–96.

doi:10.1126/science.1183439

doi:10.1371/journal.pone.0001499

 http://dx.doi.org/10.2139/ssrn.4701706

https://doi.org/10.3389/fendo.2020.577925

https://doi.org/10.3389/fendo.2020.00541

Comments on the Quality of English Language

The language of presentation needs careful correction and rewriting of the manuscript in many parts.

Reviewer 3 Report

Comments and Suggestions for Authors

My Decision on the manuscript with ID (biology-2889008) is “Major Revisions”. The authors should prepare a point-by-point response to the comments raised by the anonymous reviewer before the manuscript is considered for publication in Biology.

Q1. In the title, add Whiteleg shrimp.

Q2. Litopenaeus vannamei – should be written italic.

Q3. Table 1 and Table 2: the authors should add the following information such as NCBI GenBank accession numbers, R2, Pearson’s coefficient, efficiency, Tm, annealing temperature, product size (bp), and references related to these primers.

Q4. Line 107: The authors should add the details of the physico-chemical properties of the rearing water.

Comments on the Quality of English Language

Moderate editing of English language required

Round 2

Reviewer 2 Report

Comments and Suggestions for Authors

The authors' revision improved the article's sense considerably. Even the discussion of the results, which had previously been criticized, has significantly improved. From the point of view of science, the article is a good study; there are no problems with the methodology, and the findings support the conclusion. However, authors should double-check their text. I found numerous grammatical errors and inaccuracies. As a result, it is critical to not only accept and correct them, but also, if possible, have them read by a native speaker of the language. Overall, I enjoyed the authors' work and wished them continued success in science.

Abstract and all text should be checked and improved.

…resulting in their inability to spawn, while others undergo multiple maturations and contribute to the majority of larval supply. Despite numerous studies that

…Differences in

The expression levels… were significantly different between the two groups…

… the ovaries of high reproductive bulk at II–IV maturity stages compared to low reproductive bulk, while almost no expression was detected in the eyestalk tissue of high reproductive bulk.

The PRDM9 gene was exclusively expressed in ovarian tissue, with significantly higher expression in the ovaries of high reproductive bulk at four maturity stages compared to low reproductive bulk.

Lines 183 and 192 … subjected to

Line 203 …were reconstructed

Line 222   indicating

Line 239… with a preheated signal probe

Lines 252-253 … The maturation frequency of 632 shrimp females during a 45-day production cycle of  Pacific whiteleg shrimp is illustrated in Figure 1.

Lines 259-274: Check Grammar carefully.

between high- and low-reproductive females.

high-reproductive females

whereas the ovaries of low-reproductive females

As we progressed to stage II ..

in the ovaries of low-reproductive females continued to exhibit limited ribosome attachment

the ovaries

cristae

Subsequent to advancement

Line 300. Delete “at”.. averaging 27.71%

Line 338 …The KEGG 

Line 339-340 Escherichia coli is species Latin name/ It should be in italic

Line 384. * represents a significant difference (P<0.05), ** represents a highly significant difference (P<0.01). the difference from what or between what and what??

Line 394 of the PRDM9 gene

Line 395 the eyestalk

Line 401 the PRDM9 gene

Line 405 … high and low reproductive bulks

Line 415.. the high reproductive bulk

Line 429 and Line 434 … the high and low reproductive bulks

Line 436 in the ovary

Line 438 cristae

Line 439 mitochondrial-related genes, including the ATP synthase

Lines 440-441 … the mitochondrial fission 1 protein-like gene, the iron sulfur cluster

Lines 441-442 … the high and low reproductive bulks

Line 448 embryonic development defects in mice

Lines 452- 454 Beams et al. [34] … Jiang et al.[35]

Line 460 …the difference in mitochondria

Line 461… of the two bulks

Line 464… may be a consequence of different morphological structures of mitochondria.

Line 465 ..in the genome

Line 467-468 ..of the Plcβ4 gene in the 5-HT

Line 473 .. increasing mitochondrial DNA

Line 474 … rodent kidney

Line 479 … subsequent changes in cellular metabolism contribute to an increase in oxidative phosphorylation

Line 489 …genes that might be

Line 497 …results in further

Line 511… leads to infertility because the deletion of the PRDM9 gene

Line 523… mechanisms

Line 525… with the low- and high-fecundity shrimps

Comments on the Quality of English Language

Language correction required. I outlined in my review the main mistakes for the authors, but perhaps I missed something. Manuscript should be checked carefully.

Reviewer 3 Report

Comments and Suggestions for Authors

The authors appropriately responded to the points raised by the reviewer

Comments on the Quality of English Language

Minor editing of English language required
